# A novel probabilistic forecast system predicting anomalously warm 2018-2022 reinforcing the long-term global warming trend

Florian Sévellec [1,2] & Sybren S. Drijfhout[2,3]

In a changing climate, there is an ever-increasing societal demand for accurate and reliable interannual predictions. Accurate and reliable interannual predictions of global temperatures are key for determining the regional climate change impacts that scale with global temperature, such as precipitation extremes, severe droughts, or intense hurricane activity, for instance. However, the chaotic nature of the climate system limits prediction accuracy on such timescales. Here we develop a novel method to predict global-mean surface air temperature and sea surface temperature, based on transfer operators, which allows, by-design, probabilistic forecasts. The prediction accuracy is equivalent to operational forecasts and its reliability is high. The post-1998 global warming hiatus is well predicted. For 2018–2022, the probabilistic forecast indicates a warmer than normal period, with respect to the forced trend. This will temporarily reinforce the long-term global warming trend. The coming warm period is associated with an increased likelihood of intense to extreme temperatures. The important numerical efficiency of the method (a few hundredths of a second on a laptop) opens the possibility for real-time probabilistic predictions carried out on personal mobile devices.

[1] Laboratoire d'Océanographie Physique et Spatiale, UMR6523, Univ. Brest, CNRS-Ifremer-UBO-IRD, Brest, France. [2] Ocean and Earth Science, University of Southampton, Southampton, UK. [3] Koninklijk Nederlands Meteorologisch Instituut, De Bilt, Netherlands. Correspondence and requests for materials should be addressed to F.S. (email: florian.sevellec@univ-brest.fr)

Many studies have focused on the attribution of climate change from global to local scales[1]. These studies relate variations in observations with variations in external forcing to explain, or partially explain, the observed changes. For example, changes in global-mean surface air temperature (GMT) can be partially attributed to variations in external climatic forcing, such as volcanic eruptions or aerosol and greenhouse gas emissions[2] (Fig. 1). However, there still remains a residual to this

forced component (Fig. 1e), which can be interpreted as the internal variability of the climate. This variability, because of its dominance over the forced trend on interannual to decadal timescales (Fig. 1g), is at the heart of interannual climate prediction[3,4], and the goal of our study. Moreover, since volcanic eruptions are unpredictable by essence and aerosol and greenhouse gas emissions depend on socio-economic choices, further improvement of climate predictions will mainly occur through

**Fig. 1** Attribution of observed global-mean surface air temperature (GMT) and sea surface temperature (SST). **a, b** The total (red) annual, (purple) 5-year and (blue) 10-year variations in GMT and SST measured from 1880 are decomposed (through an attribution method based on multivariate linear regression onto volcanic eruptions, aerosol concentration, and greenhouse gas concentration[2]) into **c, d** a forced contribution and **e, f** a residual. **g, h** Relative variance of forced and residual GMT and SST changes as a function of the duration of these changes. Variations are mainly controlled by the residual, rather than forcing on interannual to decadal timescales. The observed GMT are from NASA GISS temperature data, and SST is from the NOAA ERSSTv5 record

better, more accurate predictions of the internal variability. This conclusion is also true for the global-mean sea surface temperature (SST) studied here (Fig. 1).

In this study, we predict this internal variability through the use of transfer operators trained by GMT and SST variations simulated by 10 climate models from the Coupled Model Inter-comparison Project phase 5 (CMIP5)[5]. This methods allows to determine skillful and reliable probabilistic forecasts of GMT and SST. Using this method to predict the future, the outcome is that the current climate has a large likelihood to reach a warmer than normal period over the next 5 years on top of the forced global warming trend.

## Results

**Probabilistic forecast system.** To make climate predictions we developed a PRObabilistic foreCAST system (PROCAST system) based on transfer operators. This method has been successfully used in a large range of scientific studies from statistical physics[6] to geophysical fluid mechanics[7–9]. The basic principle behind the use of transfer operators is a statistical approach to rationalize the chaotic behavior of the system. The transfer operators gather the information from all known, previous state transitions (or trajectories in the phase space), allowing the computation of the system evolution from its current state to new states in a probabilistic manner. In this sense it can be related to the analog methodologies[10–12]. Here the climate state is evaluated through the one-dimensional phase space defined by either GMT or SST, whereas the state transitions are based on GMT or SST evolutions simulated by climate models from the CMIP5 database (see Methods for further details).

The evolution of the annual GMT and SST anomalies for the 10 climate models is estimated from the temperature anomaly relative to the ensemble mean in each individual climate model ensemble (see Methods for further details). This procedure separates internal variability from the forced signal. The anomaly of GMT or SST in the observational record is computed by removing the part that can be attributed to external forcing (see Methods for further details).

The CMIP5 GMT and SST anomalies consist of centered distributions with a standard deviation of the annual mean of 0.1 and 0.07 K, respectively. For GMT, the modeled standard deviation is slightly weaker than the observed one (0.12 K), but remains in good agreement: <9% of relative difference. On the other hand, for SST, the standard deviation of the distribution in the CMIP5 is significantly weaker than in the observations (0.13 K), with a relative difference of 43%. Hence for SST, the modeled distribution is renormalized to fit the standard deviation of the observations.

To define the transfer operators, we split the phase space in 24 states and computed the probability of transition between the 24 states based on the ~10 000 transitions provided by the CMIP5 database (see Fig. 2 and Method for further details). Hence the transfer operators are a natural tool for conditional probability forecasts. Indeed, we evaluate the probability of future system states based on the condition that the system is in its current state. To gain insight on the method we applied the transfer operators to the 24 possible states. This shows that for all possible initial conditions the probability density function slowly converges to the total density distribution with a timescale of ~10 years (see Fig. 3 for an example of the asymptotic convergence). This is accomplished through the slow converge of the mean probability and the spread of the probability distribution. However, the computed probability density distribution is more intricate than simply described by the evolution of its mean and

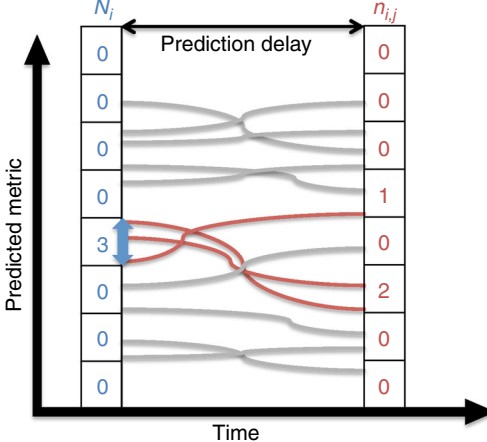

**Fig. 2** Schematic of the Transfer Operator method. The computation of the transfer operators follows four steps: first, the predicted metric (GMT and SST) is split in a number of different states based on its intensity (8 in the schematic, 24 for the real forecast system); then, the number of trajectories from the CMIP5 database in each state is counted ($N_i$, blue numbers); as a third step, for each state, the number of corresponding trajectories found after the prediction delay is counted ($n_{i,j}$, red numbers); finally, the probability of transition from $i$th state to $j$th is given by $n_{i,j}/N_i$. This sequence is repeated for 10 different prediction delays and 10 averaging times of trajectories (both are varied from 1 to 10 years, by 1 year time step) building the range of transfer operators required for prediction

standard deviation. Indeed, within the information restricted to the reduced phase space of GMT or SST, the method allows to follow the evolution of the complete probability density function over time (Fig. 3).

Hence, this method is valid under four assumptions: GMT or SST information is enough for GMT or SST prediction; all model trajectories are statistically equivalent; the common component of each model reflects its forced component; and stationarity of the statistics of anomalies in a changing climate scenario (these assumptions are further described in the Methods). In particular, the severe truncation of the phase space to a single variable implies that different climate states with equivalent GMT or SST are all aggregated in the probabilistic approach of the transfer operators. However, as it will be fully described later, PROCAST is skillful for interannual GMT and SST prediction despite these four assumptions. This, a posteriori, confirms that despite these assumptions not being strictly true, they are reasonable given the aims of our study.

**Evaluation metrics and perfect model approach.** Now that the transfer operators are defined and its general behavior described, we first estimate the predictive skill of our method in a perfect model framework. This means that we predict trajectories of the numerical models, rather than of real observations. This allows to estimate the predictive skill in the best possible scenario, since it avoids the intrinsic bias or error between models and observations. The probabilistic distribution is computed for up to 10-year lags (1–10 years ahead) and for annual to 10-year averaged data.

To estimate the validity of our probabilistic predictions we use two different measures: the coefficient of determination—$R^2$, which shows the skill of the mean prediction; and the reliability, which measures the accuracy of the spread in the prediction.

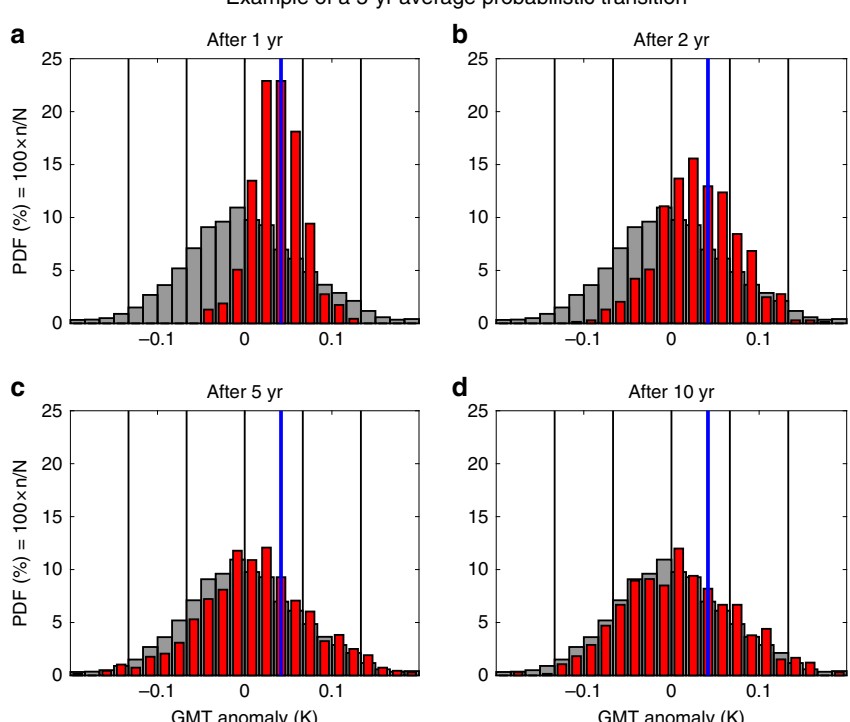

**Fig. 3** Example of a probabilistic interannual prediction for 5-year GMT anomalies using the transfer operators. Evolution of the predicted density distribution of GMT for hindcast lags of **a** 1 year, **b** 2 years, **c** 5 years, and **d** 10 years (red histogram). Vertical blue lines indicate the initial condition; the gray histograms in the background represent the asymptotic, climatological distribution; vertical black lines correspond to the mean, ±1, and ±2 standard deviations of the asymptotic, climatological distribution

These two measures can be mathematically expressed as:

$$R^2 = 1 - \frac{\overline{\left(\overline{x_i p_i(t)}^i - o(t)\right)^2}^t}{\overline{o(t)^2}^t},$$

(1)

$$\text{Reliability} = \sqrt{\overline{\left\{ \frac{\overline{\left(\overline{x_i p_i(t)}^i - o(t)\right)^2}^t}{\overline{\left[x_i - \overline{x_i p_i(t)}^i\right]^2 p_i(t)}^i} \right\}}},$$

(2)

where $t$ is time, $i$ is the possibility or state index, $o(t)$ is the observation, and $x_i$ are the predicted possibilities with probability $p_i(t)$. The bar denotes an average over time or sum over possibilities depending on the superscript. (Our equation of the reliability is an extension for non-stationary statistic of the previously suggested definition[13].) The coefficient of determination, when multiplied by 100, gives the percentage of variance of the observations explained by the prediction. Since the system is chaotic (there is a degree of uncertainty around the mean prediction), it is expected that the prediction cannot represent the observation perfectly, even if the model represents perfectly reality. Hence, the reliability measures the accuracy of this prediction error. When a reliable prediction has large skill (~1) we expect the prediction uncertainty to be small. On the other hand, when a reliable prediction system has low skill (~0) we expect the prediction uncertainty to be as big as the observed variance. In this context, and regardless of its skill, a reliable prediction system always needs to have a reliability close to 1. Hence, despite that a high value of $R^2$ is preferable for a skillful prediction, the reliability is arguably more important to estimate

the usefulness of the prediction system. Indeed, a reliable prediction system can be used for probabilistic forecasts and risk assessments, even if it has low skill[14,15].

To evaluate the quality of our prediction system, skill and reliability are computed for all lags and averaging times. For comparison, we use persistence as our null hypothesis (i.e., initial values in our hindcast for all hindcast times). Within the perfect model approach, all the trajectories in the selected models of the CMIP5 database have been tested. This reveals two main results: PROCAST is able to surpass persistence for all averaging timescales and hindcast lags for GMT and SST; and prediction skills are often larger than 0.5 on annual to interannual timescales (Fig. 4). It also reveals the excellent reliability of PROCAST (as expected within a perfect model approach) with a value of 1 (within a 6% and 3% errors for GMT and SST, respectively) for all averaging timescales and hindcast lags (Fig. 4c, d).

To further test the predictive skill and reliability of PROCAST we have assessed them in an imperfect model approach (i.e., removing outputs of one model from the transfer operators computation and using them as pseudo-observations). We find that PROCAST is still able to perform at the same level of accuracy than within the perfect model approach with a slight decrease of the coefficient of determination of less than 0.01 for all lags and averaging times tested.

**Hindcast skills and predictions of the post-1998 hiatus.** After having tested PROCAST in a perfect model setting, we now test the exact same system with real observations. (Note that no retuning before going to observations has been applied.) We reproduce the skill analysis with the observed internal variability, estimated as anomalies from the forced component in GMT and SST (Fig. 1). For this purpose we computed retrospective predictions of the past, or hindcasts, from 1880 to 2016. This

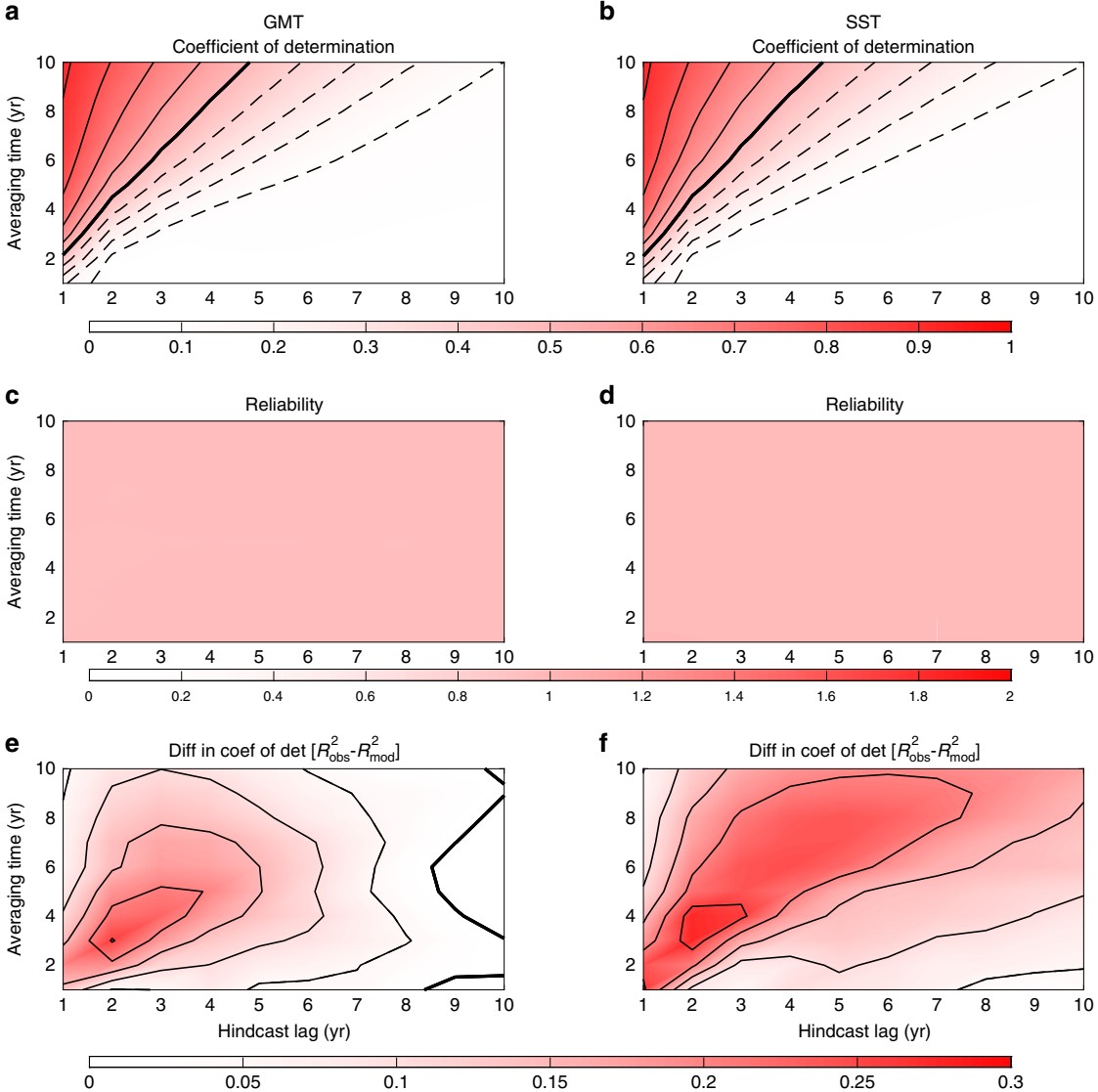

**Fig. 4** Interannual hindcast skills of GMT and SST within a perfect model approach. **a**, **b** Skill of the prediction measured by the coefficient of determination —$R^2$—between model observations and mean prediction for different hindcast lags and averaging times. The coefficient of determination between model observations and persistence (i.e., the null hypothesis of prediction) is also computed to give a benchmark. Thick contour lines represent values of 0.5, thin dashed—lower skill, and thin solid—higher skill, with contour intervals of 0.1; and hatching shows skill lower than the persistence. **c**, **d** Reliability of the prediction for different hindcast lags and averaging times. Note the absence of hatched region in **a** and **b** denoting the better skill than persistence for all hindcast lags and averaging times. Also, note the flat pink color in **c** and **d** corresponding to a good reliability close to 1 for all hindcast lags and averaging times, as expected in a perfect model approach. **e**, **f** Difference between the coefficient of determination for the hindcasts with observations and within a perfect model approach (**a** and **b** vs. Fig. 5c, d for GMT and SST). For all hindcast lags and averaging times, the skill is better for observations than for the perfect model approach. Thick black lines are for zero values, thin black lines are positive values with a contour interval of 0.05

procedure allows a full estimate of the predictive ability of our prediction system in the most realistic and operational setting. (Examples of hindcasts for 5-year averages are shown in Fig. 5a, b.)

As before, the probabilistic distribution is computed for up to 10-year lags and for annual to 10-year averaged data. Similar to the perfect model approach, skill values decrease from 1 to 0 depending on the lags (Fig. 5). Focusing on a 5-year mean prediction of SST, we still have a skill for 5-year lags of ~30% (Fig. 5d), suggesting our ability to accurately forecast part of the SST variations for the next 5 years. More generally the skills are always better than persistence (except for a few averaging times and hindcast lags of SST). Also reliability remains close to 1, even when the skill is low, suggesting that even for low skills we are sampling an accurate range of possible future states. The same

holds for predictions of GMT, but with slightly reduced skill compared to SST. This demonstrates the usefulness of PROCAST for probabilistic predictions.

The root mean square error averaged for hindcast lags from 1 to 9 years for annual GMT is 0.105 K. This is strictly identical to the value reported for DePreSys in 2007 (the operational Decadal Prediction System of the Met-Office)[16]. When averaged over the hindcast lags from 1 to 5 years, the root mean square error of PROCAST is 0.104 K, whereas it is 0.151 K for the latest version of DePresys (DePreSys3, D. Smith, personal communication). This indicates that PROCAST is 37% more accurate than DePreSys3 for interannual predictions of GMT. This comparison is slightly biased, however, since DePreSys predicts the absolute temperature, whereas we only predict the anomaly from the forced part. However, the forced part of the variability is arguably

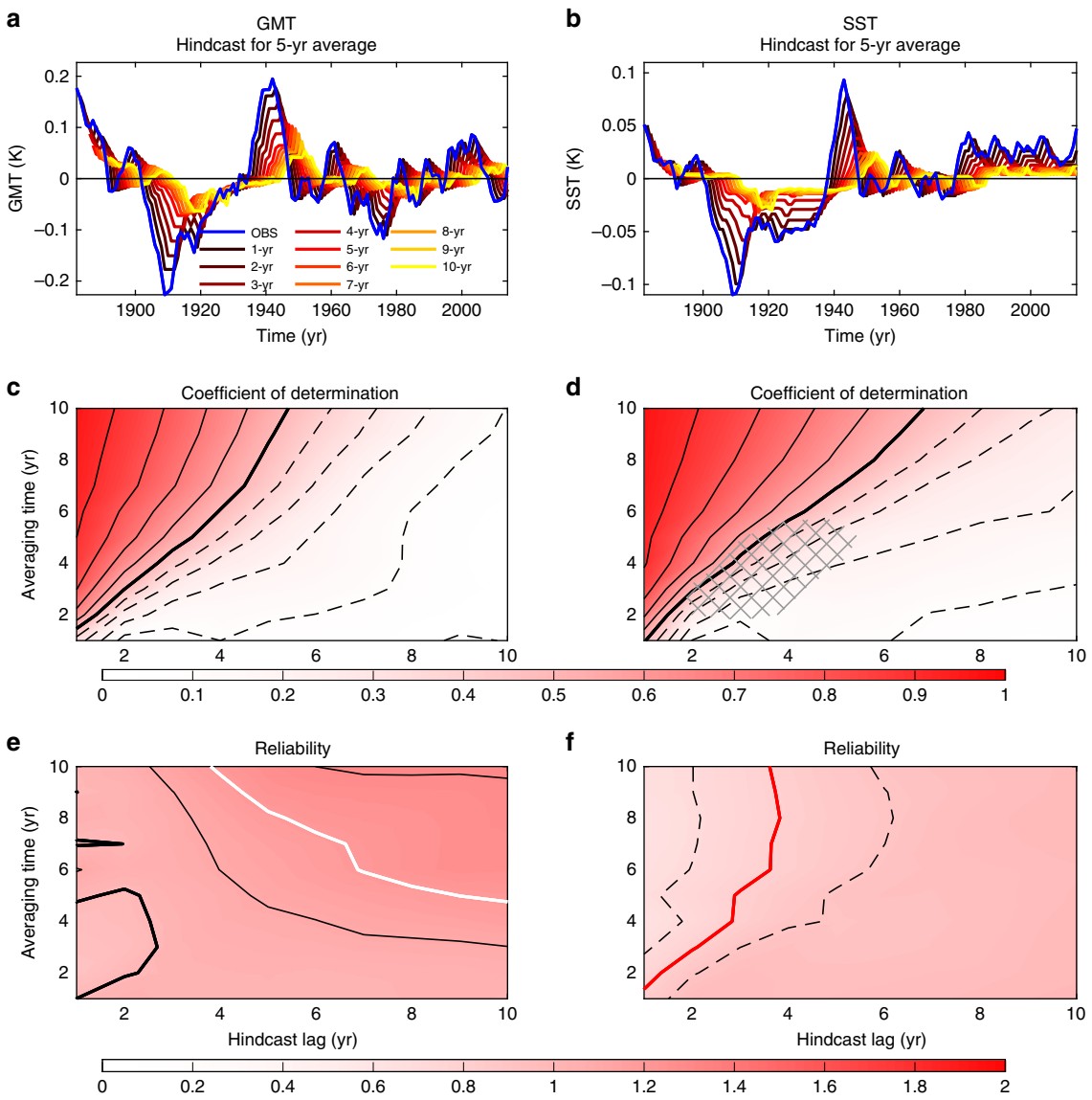

**Fig. 5** Interannual hindcast skills of the observed GMT and SST. Example for 5-year average data of **a**, **b** GMT and SST observations (blue) with mean prediction for 1- to 10-year hindcast lags (from black to yellow lines, respectively). **c**, **d** Skill of the prediction measured by the coefficient of determination—$R^2$—between observations and mean prediction for different hindcast lags and averaging times. The coefficient of determination between model observations and persistence (i.e., the null hypothesis of prediction) is also computed to give a benchmark. Thick contour lines represent values of 0.5, thin dashed—lower skill, and thin solid—higher skill, with contour intervals of 0.1; and hatching shows skill lower than the persistence. **e**, **f** Reliability of the prediction for different hindcast lags and averaging times. Thick red, black, and white contour lines represent values of 0.8, 1, and 1.2, respectively; thin black dashed and solid lines represent lower and higher values than 1, respectively, with contour intervals of 0.1. Note the better skill than persistence (almost no hatched region) and the good reliability close to 1 for all hindcast lags and averaging times

the most predictable, since the external forcing is imposed accurately during hindcasts. Furthermore, the accurate probabilistic approach of PROCAST also differs from DePreSys, which can only diagnose a probabilistic prediction from a relatively small ensemble (~10 members), limiting its statistical accuracy and overall reliability. More specifically, PROCAST reliability for annual GMT is almost perfect with an averaged value for hindcast lags from 1 to 5 years of 1 (prediction spread is as big as the prediction error on average), whereas it is 2.3 for DePreSys3 (prediction spread is 2.3 times as small as the prediction error on average). This relatively weak reliability of DePreSys3 is a sign of the under-dispersion of the ensemble in comparison with its prediction error and of an over-confident prediction system. Hence, PROCAST appears to be better suited for probabilistic predictions and risk assessments of extremes. Finally, and

probably more importantly, the numerical cost is without comparison. Doing a 10-year forecast using PROCAST takes 22 ms, and can be easily done on a laptop almost instantaneously. On the other hand, a 10-year forecast using DePreSys3 (which corresponds to a 10-member ensemble) takes a week on the Met-Office supercomputer, accessible only to a small number of scientists. This difference in numerical cost has to be put into perspective, though. PROCAST takes advantage of the freely available CMIP5 database, which is an incredibly expensive numerical exercise of the worldwide climate science community. Also, unlike PROCAST, DePreSys3 is not specifically trained for a single-variable prediction, so that the entire climate state is predicted in one forecast. This is obviously beneficial.

To further identify the usefulness of PROCAST we tested its prediction skill for the recent global warming hiatus[17]. We define

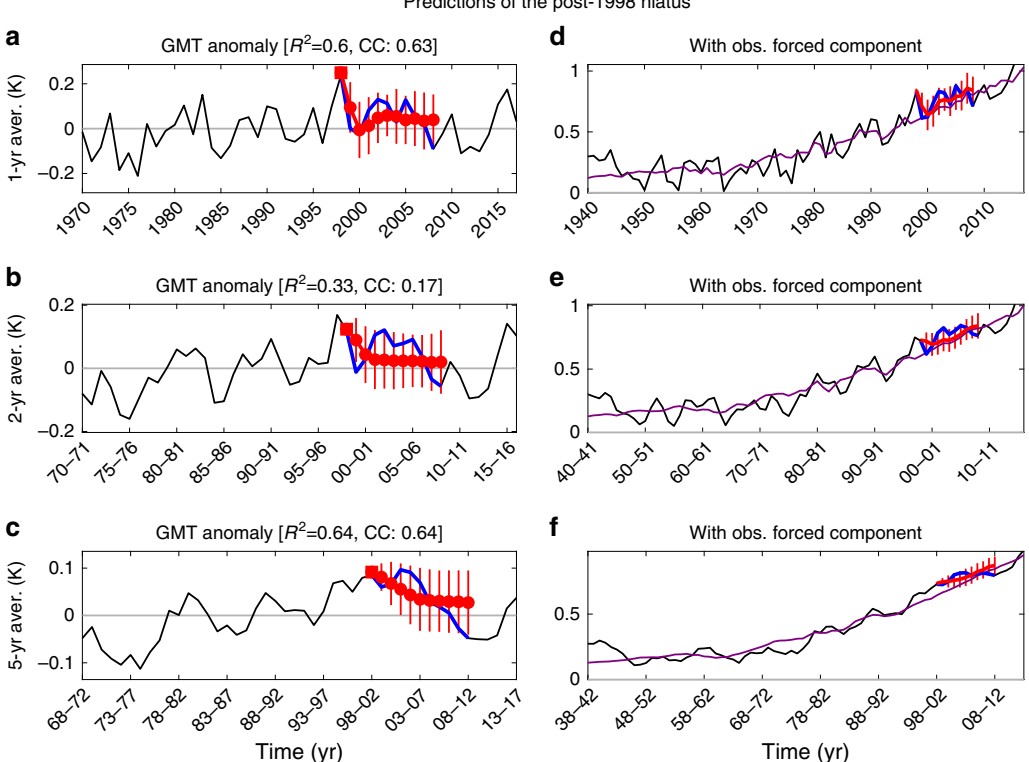

**Fig. 6** Predictions of the post-1998 hiatus observed through GMT anomalies. Observation and prediction of GMT anomalies averaged over **a** 1 year, **b** 2 years, and **c** 5 years. Hiatus is defined as the post-1998 decade (blue line) showing a decrease of GMT anomalies partially or totally offsetting the forced trend. Decadal prediction means and standard deviations (red circles and vertical lines, respectively) are obtained using PROCAST initialized in 1998 (red squares). Prediction skills are computed through the coefficient of determination ($R^2$) and the correlation coefficient (CC) over the post-1998 decade. **d**–**f** are equivalent to **a**–**c** with the addition of the observed component attributed to forcing (purple lines)

this recent hiatus as the post-1998 decade cooling seen in GMT anomaly (Fig. 1e). This cooling totally offsets the forced warming (Fig. 1c) leading to a plateau in the observed, total warming (Fig. 1a). PROCAST is indeed able to reproduce the decade-long cooling anomaly (Fig. 6) for all averaging timescales tested (1–10 years). The coefficient of determination is 0.52 on average (for averaging timescales ranging from 1 to 5 years and could be as high as 0.6 or 0.64 for 1 and 5-year time averages, respectively. This suggests that 60% and 64% of the annual and 5-year variations, respectively, are accurately predicted for a decade long. For these two examples the correlation coefficient is also high with values of 0.63 and 0.64. Despite some error in its exact intensity (especially when focusing on mean prediction) or the details of its annual variations, this shows that an event such as the post-1998 hiatus could not have been missed using PROCAST (especially when acknowledging the predictive spread). In particular our probabilistic forecast framework shows that a decade-long hiatus was always a likely outcome (always well within 1 standard deviation of our prediction), even if not the most likely, especially after 7 years. Because the amplitude is somewhat lower than observed, it would be consistent if a small part of the hiatus was indeed caused by external forcing, although the main part would be due to internal variability[18–21]. This is a significant achievement since the recent hiatus can be considered as a statistical outlier[22,23], and only a few of the CMIP5 models simulated such a strong and long pause in global warming[24]. This places PROCAST among the state-of-the-art prediction systems, which have been able to retrospectively predict the recent global warming hiatus[25,26]. Other starting dates have been tested (such as 2002) and always allow PROCAST to capture the long-term

hiatus, but to a lesser degree the exact interannual variation of the decades.

When compared with the perfect model predictions, the predictive skill in PROCAST is (always) better for real-world predictions (Fig. 4e, f) than in the perfect model approach. In particular, the skill is improved by up to 30% on interannual timescales, and is especially better for SST. This behavior has been previously reported for other variables and prediction systems[27–29] and seems to be related to a weaker signal-to-noise ratio in models than in observations.

It is also interesting to note that skill and reliability are improved by the addition of information from more models rather than by selecting a subset of the best models (i.e., models giving the best skill when used alone to train the Transfer Operator). This means that the transfer operators built with only the best models have lower skill than the transfer operators built with all 10 climate models. Moreover, removing any single model from the set of 10 does not significantly lower the skill, suggesting that convergence has been reached when using 10 climate models.

Our analysis also shows that SST has better skill than GMT for all tested hindcast lags and averaging times. This suggests that the ocean is improving the hindcast skill and is more predictable than the continental surface temperature encompassed in GMT. This result is consistent with previous analyses suggesting the ocean as a source of predictability and limiting the continental predictability to marine-influenced regions[30,31].

**Forecasting the future**. After the skill of the method has been assessed, we compute a probabilistic forecast of GMT and SST from 2018 to 2027 (Figs. 7 and 8), focusing on three averaging

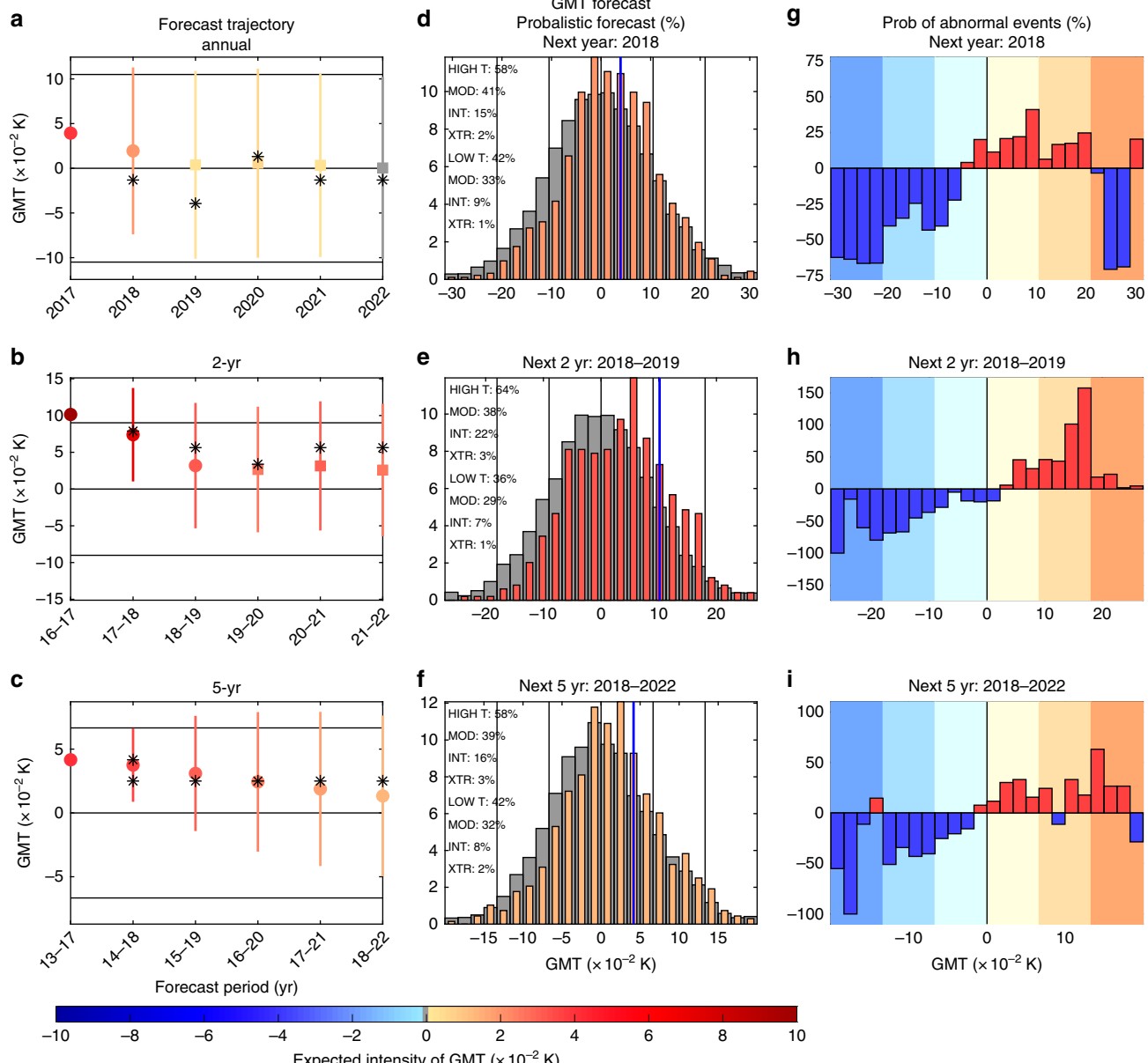

**Fig. 7** Interannual probabilistic forecast of the GMT anomalies. **a–c** Mean prediction of GMT anomaly for the next 5 years for three different averaging times: annual, 2, and 5 years; horizontal black lines correspond to the mean and ±1 standard deviations of the climatological distribution. Circles and squares represent mean predictions with coefficient of determination bigger and smaller than 0.2 (note that we still have good reliability even for skills smaller than 0.2); colorscale represents the mean prediction of the GMT anomaly; the vertical colored lines represent ±1 standard deviation of prediction distributions. Stars denote the most likely state from the distributions. **d–f** Prediction of GMT distribution (in %) for 1, 2, and 5 years in advance with respect to 2017. Gray histograms in the background represent the asymptotic, climatological distribution; vertical blue lines represent the current position used to initialize the forecast system; vertical black lines correspond to the mean, ±1, and ±2 standard deviations of the climatological distribution. **g–i** Distribution of probability anomaly (in %), probability changes with respect to the climatological distribution, for the 1, 2, and 5 years in advance predictions. Background colorscale represents ±0–1, 1–2, and more than 2 standard deviations, respectively, consistently with moderate, intense, and extreme events

times: 1-, 2-, and 5-year averages. We also considered longer averaging times, such as a 10-year averaging period, but these all show forecast probabilities almost equivalent to the climatological probability (Table 1). Similarly, for long enough lags or forecast times (i.e., a decade) the predictions always converge to the climatological values (Figs. 7 and 8a–c).

With shorter time averages our probabilistic forecast for 2018 (based on data up to 2017) suggests a higher likelihood of warm events for both GMT and SST (Figs. 7 and 8d), with a probability of higher temperature than predicted by the forcing alone of 58% and 75% for GMT and SST, respectively. This corresponds to an

expected warm anomaly of 0.02 and 0.07 K for GMT and SST, which would reinforce the forced warm trend. To describe the expected warm event in greater details, we can classify the temperature anomaly as moderate (lower than 1 standard deviation), intense (bigger than 1 standard deviation and lower than 2 standard deviations), and extreme (bigger than 2 standard deviations). This classification suggests that moderate warm events are the most likely for 2018 GMT and SST. This can be further diagnosed by looking at the relative changes of probabilities from the climatological probability (Figs. 7 and 8g). This suggests that intense and extreme cold events have the

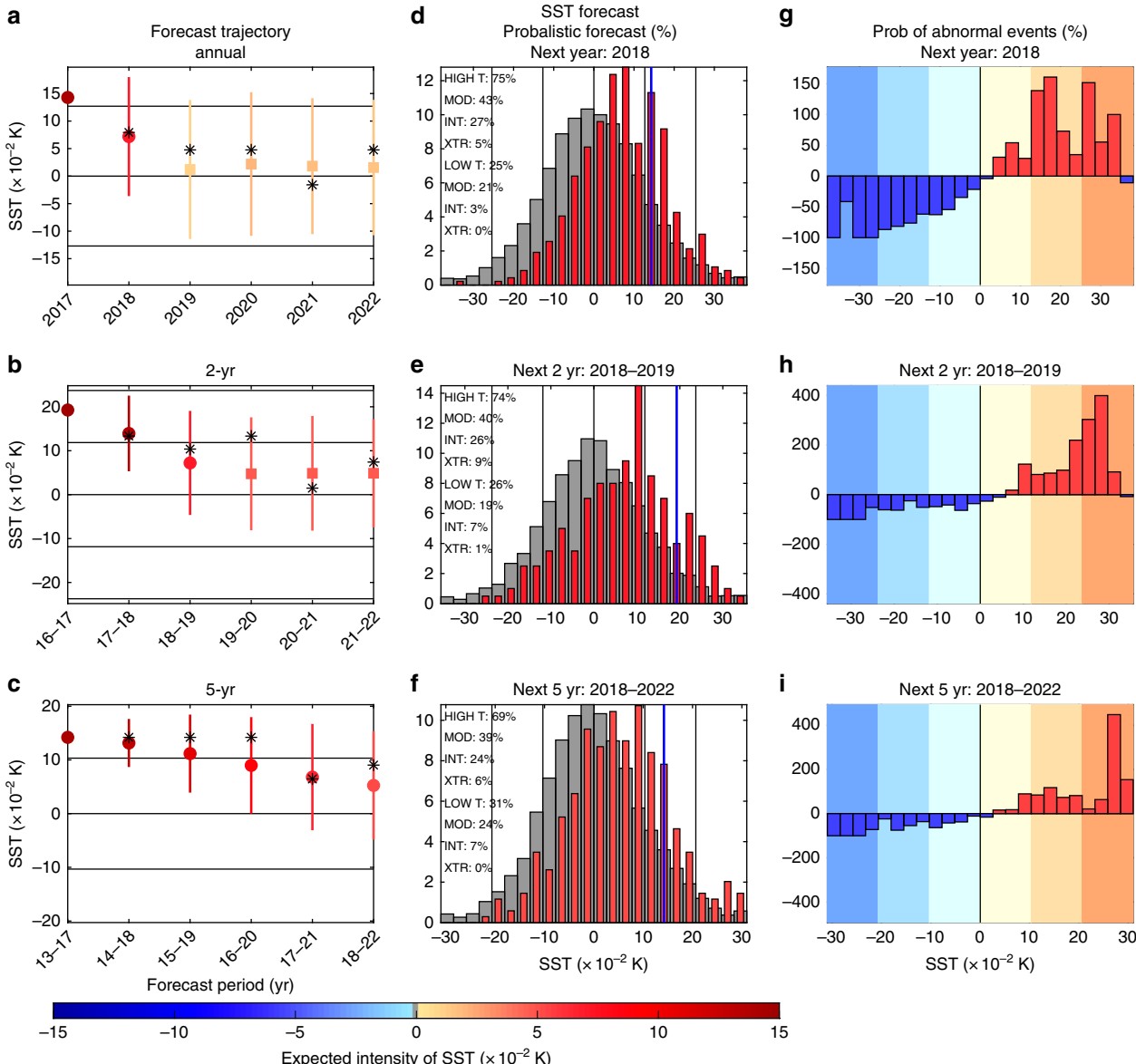

**Fig. 8** Interannual probabilistic forecast of the SST anomalies. **a–c** Mean prediction of SST anomaly for the next 5 years for three different averaging times: annual, 2, and 5 years; horizontal black lines correspond to the mean, ±1, and ±2 standard deviations of the climatological distribution. Circles and squares represent mean predictions with coefficient of determination bigger and smaller than 0.2 (note that we still have good reliability even for skills smaller than 0.2); colorscale represents the mean prediction of the SST anomaly; the vertical colored lines represent ±1 standard deviation of prediction distributions. Stars denote the most likely state from the distributions. **d–f** Prediction of SST distribution (in %) for 1, 2, and 5 years in advance with respect to 2017. Gray histograms in the background represent the asymptotic, climatological distribution; vertical blue lines represent the current position used to initialize the forecast system; vertical black lines correspond to the mean, ±1, and ±2 standard deviations of the climatological distribution. **g–i** Distribution of probability anomaly (in %), probability changes with respect to the climatological distribution, for the 1, 2, and 5 years in advance predictions. Background colorscale represents ±0–1, 1–2, and more than 2 standard deviations, respectively, consistently with moderate, intense, and extreme events

lowest risk in 2018 (i.e., largest change of occurrence compared to climatology), with an occurrence decrease of more than 60% and 100% for GMT and SST, respectively.

On longer timescales, for the 2018–2019 average, anomalous warm events remain the most likely event (Figs. 7 and 8e), with an expected intensity of 0.03 and 0.07 K for GMT and SST. For this period, intense GMT warm anomalies show the maximum changes in likelihood compared to climatology (Fig. 7h). For SST the maximum change in likelihood is an increase of extreme warm events (Fig. 8h).

For the forecasted 5-year averaged temperatures (i.e., for the period 2018–2022), the predictions differ. For GMT, the predictions suggest a balanced probability between warm and

cold events (Fig. 7f) on top of the forced trend, with a small relative reduction of expected occurrence of extreme cold and a small relative increase of expected occurance of extreme warm events (Fig. 7i). For SST, the forecast still suggests an anomalous warm event for the period 2018–2022 with an expected value of 0.05 K (Fig. 8f). The forecast also suggests a relative increase of the probability of extreme warm events for SST over this period, by up to 400% (Fig. 8i).

## Discussion

It is now well understood that global warming is not a smooth monotonous process[32]. Variations around the continuous

warming can even dominate the trend on decadal timescales[23,25,33], as was the case for the hiatus event in the early twenty-first century[17,18]. In this study we used CMIP5 simulations to train a statistical model to predict variations of GMT and SST with respect to the forced trend on interannual to decadal timescales. The statistical model is based on the Transfer Operator framework that transforms deterministic trajectories into probabilistic ones. Hence, our derived prediction system is naturally fitted for probabilistic forecasts, and is named PROCAST.

For both metrics we show that our prediction system is able to be more accurate than persistence within a perfect model approach. We also showed the ability of PROCAST to be reliable even when skills are low, suggesting the usefulness of the predictive system and of its probabilistic approach on interannual to decadal timescales.

To further identify the accuracy of PROCAST, we computed a range of historical hindcasts from 1880 to 2016 of the GMT and SST anomalies (defined as the residuals after removing the forced components[2]). For the whole period, and with a start date every year, we evaluated the predictive skill and reliability for 1- to 10-year lags and for annual to decadal variations. This evaluation reveals that the prediction skills (measured through the coefficient of determination) outperformed the skills obtained within a perfect model approach. In particular, a retrospective prediction using PROCAST is able to capture the decade-long post-1998 global warming hiatus (Fig. 6).

Beyond the predictive skills, the reliability of PROCAST is high, suggesting that PROCAST is also able to predict the possible range of GMT and SST anomalies with their associated probability, making it well fit for probabilistic forecasts and risk assessment. For example, the post-1998 hiatus has been shown to be a likely outcome of PROCAST, despite being considered as a statistical outlier[22,23]. Also, the high reliability suggests that the 10 climate models used to train PROCAST represent accurately the statistics of the observations. Despite intrinsic limitations and biases, this reinforces the high potential of climate models for understanding the climate system.

Using our novel forecast system, we made interannual predictions for the future. These predictions suggest that 2018 has a high probability of having a warm anomaly (58% and 75%) compared to the forced trend, with expected anomalies of 0.02 and 0.07 K for GMT and SST, respectively. This occurs through the significant decrease of likelihood of extreme cold events (cold events bigger than 2 standard deviations). For the next 2 years, both GMT and SST suggest a likelihood of warm events of more than 64% and 74%, respectively. This is mostly due to an increase of the likelihood of intense warm events (between 1 and 2 standard deviations) for GMT and of extreme warm events for SST. On even longer timescales, GMT suggests an almost perfectly balanced probability between warm and cold events, whereas SST suggests a higher probability of warm events of 69% for 2018–2022, with a dramatic increase of up to 400% for an extreme warm event likelihood.

Overall, PROCAST suggests that the current warm anomaly recorded in GMT and SST is expected to continue for up to the next 5 years (Table 1), and even possibly for longer for SST. PROCAST shows better skill than DePreSys3 (the latest version of the operational Decadal Prediction System of the Met-Office), with extremely accurate reliability. Whereas classical forecast systems give the entire climate state in a single prediction, but are numerically costly (need of supercomputers), PROCAST is extremely efficient numerically but forecasts only a single metric. However, this efficient system can be easily transferred to predict other relevant changes of the climate system, such as precipitation, and to focus on regional scales more aligned with societal demand. This also opens the possibility of giving access to climate forecast, and possible subsequent regional climate impacts that scale with GMT or SST (such as precipitation extremes[34], severe droughts[35], or intense hurricane activity[36], for instance), to a wider scientific community (without the need for supercomputer) and to the general public by running a simple application on a personal portable device.

## Methods

**Computation of transfer operators for GMT and SST anomalies.** The statistics needed to develop the transfer operators[6–9] followed a multimodel approach using 10 climate models. The GMT and SST data are estimated based on historical simulations followed by the Representative Concentration Pathway 8.5 simulations. These simulations were gathered from the CMIP5 database[5]. The 10 models are (with the number of ensemble members used in square brackets): "CCSM4" [6]; "CNRM-CM5" [5]; "CSIRO-Mk3-6-0" [10]; "CanESM2" [5]; "HadGEM2-ES" [3]; "IPSL-CM5A-LR" [4]; "FIO-ESM" [3]; "MPI-ESM-LR" [3]; "MIROC5" [3]; and "EC-EARTH" [16]. These models have been selected from the CMIP5 database because they have at least three members and the required data fields. For each model, to obtain the trajectory anomalies, the multi-member mean is removed from individual trajectories. Finally, for the purposes of this study GMT and SST are time averaged using a simple running average set to $T = 1, 2, 3, 4, 5, 6, 7, 8, 9,$ and 10 years.

To determine the transfer operators we split the one-dimensional phase space defined by GMT or SST with a uniform resolution of $\eta$. Individual grid box lengths are $6\sigma/\eta$ where $\sigma$ is the standard deviation of GMT or SST. (The most extremes boxes reached infinity to cover the entire phase space, including extreme cases.) We set $\eta = 24$ for numerical applications. This number allows a good balance between high resolution (number of boxes) and reliable statistics (number of transition in each boxes). With these numbers the Transfer Operator is able to represent 24 different states with 500 individual transitions on average to build an accurate statistical transition between states.

Further increase of the number of states does not show any improvement in the accuracy of the forecast system in terms of skill nor reliability (as evaluated through last century hindcasts). Considerations to use a two-dimensional (2D; by using the first-time derivative) and three-dimensional (3D; by using both first- and second-time derivatives) phase spaces to define the transfer operators have also been given, but did not show any improvement of the method.

It is fundamental to note that we severely reduced the phase space of the climate dynamics by considering a one-dimensional phase space defined by GMT or SST. In this context, different climate states with equivalent GMT or SST are simply aggregated in the probabilistic approach of our statistical model. Arguably, for 2D or 3D variables this approach is invalid without further adjustments, as the models fundamentally differ in background state and patterns of variability. For globally averaged variables, like temperature, this is much less the case, as regional atmospheric and ocean dynamics have much less impact on these variables and their evolution is more governed by thermodynamics common to all models. As a result, the multimodel mean ensemble is often used for best guesses of historical evolution and future projections[32]. The skillfulness of PROCAST suggests that such severe truncation indeed allows accurate and reliable prediction of global mean temperature.

The transfer operators are built by evaluating the number of trajectories from the entire multimodel database in each state (individual grid box) and then evaluating the number of these trajectories ending-up in each possible state after a given transition time ($\tau$). The ratio of these two numbers gives the probability from a trajectory in an initial state to end-up in a final state after a time $\tau$ (Fig. 2). The probability of state transition is repeated for $\tau = 1, 2, 3, 4, 5, 6, 7, 8, 9,$ and 10 years, leading to 10 transfer operators for each of the 10 averaging times $T$ (so a total of 100 transfer operators).

Hence, this method allows to propagate any probability density function forward in time. Examples of statistical transitions starting from current GMT (i.e., values of 2013–2017) for the 5-year average trajectories ($T = 5$ years) and for transition timescale of $\tau = 1, 2, 5,$ and 10 years are given in Fig. 3.

The full computation of all the requested transfer operators on a typical laptop takes ~30 s and can be saved for future use.

It is crucial to note that we do not apply a single Transfer Operator successively (e.g., applying the 1-year Transfer Operator twice to get a 2-year prediction). In contrast, we apply a range of transfer operators sequentially (i.e., applying the 1-year Transfer Operator for 1-year prediction, applying the 2-year Transfer Operator for 2 years, and so on). The essential difference with the traditional use of transfer operators (i.e., applied successively as a Markovian chain) is that we do not require that the 2-year Transfer Operator is equal to applying the 1-year Transfer Operator twice successively. So, we need to, and did, establish all different transfer operators for different lags separately and independently. This method indeed allows us to avoid the need for a Markovian chain (and its required properties), which, as a result, is not verified in our reduced single-variable space[6], and is probably not valid either[37]. It should be emphasized that our method is closer to a conditional probabilistic prediction (where the condition is based on the current

| Table 1 Likelihood of anomalously warm years for the next decade | | |
|---|---|---|
| **Forecast period** | **GMT** | **SST** |
| 1 year: 2018 | 58% | 75% |
| 2 years: 2018–2019 | 64% | 74% |
| 3 years: 2018–2020 | 70% | 71% |
| 4 years: 2018–2021 | 72% | 69% |
| 5 years: 2018–2022 | 58% | 69% |
| 6 years: 2018–2023 | 55% | 65% |
| 7 years: 2018–2024 | 45% | 58% |
| 8 years: 2018–2025 | 50% | 59% |
| 9 years: 2018–2026 | 49% | 60% |
| 10 years: 2018–2027 | 50% | 62% |

GMT or SST) than to the traditional use of Transfer Operator within a Markovian chain.

As described in the main text, our method is valid under a few assumptions: first, GMT or SST information is enough for GMT or SST prediction; second, all model trajectories are statistically equivalent; third, the common component of each model reflects its forced component; and fourth, stationarity of the statistics of anomalies in a changing climate scenario (i.e., non-autonomous system). With respect to the first assumption, we acknowledge that further refinement is possible, however, as will be demonstrated, using only globally averaged SAT and SST information is already enough to arrive at a skillful prediction. Assumption 2: Model trajectories are in general not statistically equivalent and maybe severely biased with respect to the real world. For globally averaged temperature, however, the statistical differences are small, and biases for the SST ensemble as a whole have been remedied by scaling the standard deviation. (Rescaling of individual SST trajectories to observation variability does not show improvement in prediction accuracy and shows lost of reliability in comparison with the global ensemble rescaling.) Assumption 3 is clearly incorrect, as long as the ensemble size of each model is finite. However, the error made by this assumption is small enough to allow skillful prediction. Assumption 4 is also not correct, although generally applied in climate science. The general assumption is that the perturbation implied by global warming is too small to fundamentally change the anomalies, or climate variability (i.e., anomalies and perturbation to the background do not affect each other). This might be problematic when studying extreme events, whose occurrences might change in a changing climate. Hence, this assumption is questionable for certain variables that are subject to order one changes under climate change, such as, e.g., the Atlantic Meridional Overturning Circulation, or extremes in the tail of a distribution. However, there is no indication that the assumption is not approximately valid for GMT, whose variability is for a large part dominated by El Niño-Southern Oscillation and the Interdecadal Pacific Oscillation. So, despite these various assumptions, PROCAST is skillful for interannual GMT and (globally averaged) SST prediction, proving that these assumptions although not strictly true are reasonable in the context of our study, because they apply in most cases of variation in GMT and SST.

**Computation of internal variability of observed GMT and SST**. Observed GMT and SST are computed as spatial averages (NASA GISS temperature record for GMT; the NOAA ERSSTv5 record for SST), where spatial gaps in the data are ignored. The percentage of missing data (important before 1958) does not show any impact on the prediction skill. The internal variations in GMT and SST in the observational record are computed as the residual after having removed the part that can be attributed to external forcing. This attribution is based on a multivariate linear regression onto volcanic eruptions and greenhouse gases and aerosol concentrations[2]. Hence, for removing the part attributed to external forcing from the time series of SAT and SST a multiple linear regression analysis is performed. The approach assumes that globally averaged temperature responds linearly, with some lag, to the various forcing agents. After removing the average value (we are only interested in anomalies), we can write:

$$V(t) = \varepsilon + \Sigma_{i=1}^{k} a_i F_i(t - l_i), \qquad (3)$$

where $t$ is the time, $V$ is the total GMT or SST, $k$ is the number of forcings considered (i.e., three: greenhouse gases, aerosol concentrations, and volcanoes), $a_i$ is the regression coefficient, $F_i$ is the forcing time series, $l_i$ is the lag by which temperature responds to the forcing, and $\varepsilon$ is the residual. The forcing time series are taken from the CMIP5 historical plus RCP4.5 (after 2005 till present) forcing dataset. We refer to ref. [2] for more details and figures of how the regression performs.

We also considered subtracting the impact of solar forcing, but have disregarded this for three reasons. First, direct measurements start only in 1978. If we limit ourselves to this period the length of the time series is shortened and thereby limits the robustness of any computed prediction skill. Second, for longer time series solar forcing is based on a reconstruction, which appears to be unstable

leading to large unrealistic trends degrading the prediction skill. Third, solar irradiance is weak compared to other forcing and internal variability[2]. Hence, including it in the internal variability hardly affects the results.

**Model availability**. The model developed and used during the current study are available from the corresponding author on request.

**Data availability**. The datasets analyzed and/or generated during the current study are available from the CMIP5 and KNMI Climate Explorer webpage and/or from the corresponding author on request.

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

## Acknowledgements

This research was supported by the Natural and Environmental Research Council UK (SMURPHS, NE/N005767/1), by the SYCLOPE project funded by the Southampton Marine and Maritime Institute (University of Southampton), and by the DECLIC and Meso-Var-Clim projects funded through the French CNRS/INSU/LEFE program. The authors acknowledge the World Climate Research Programme's Working Group on Coupled Modelling, which is responsible for CMIP, and we thank the climate modeling groups for producing and making available their model output (listed in the Methods of this paper). For CMIP the U.S. Department of Energy's Program for Climate Model Diagnosis and Intercomparison provides coordinating support and led development of software infrastructure in partnership with the Global Organization for Earth System Science Portals.

## Author contributions

The two authors contributed equally to the experimental design and the writing of the manuscript. F.S. computed the transfer operators and ran the experiments of PROCAST. S.S.D. did the attribution analyses of observations and of CMIP5 data.

## Additional information

**Competing interests:** The authors declare no competing interests.

