## [Peer Review File · Nature Communications]

Editorial Note: Parts of this peer review file have been redacted as comments were included that were not part of the transparent peer review scheme.

Reviewers' comments:

Reviewer #1 (Remarks to the Author):

Dear authors,

let me (Valerio Lucarini) say that I honestly appreciate the style and goals of your paper (at many levels), but I am afraid that I have to give a negative evaluation of its content for a couple of reasons.

The more important reason has to do with the interpretation of the transfer operator in the case of reducing the system to just one variable - globally averaged surface temperature or sea temperature.

The transfer operator is "well defined" if the system one is considering obeys the semigroup property. This means - broadly speaking - that the operator describing the transition of probabilities between 0 and time t can be written as the product of the evolution operator between 0 and $t-\tau$ and the one between $t-\tau$ and t , for any choice of τ . This is the case if one works in the full space of the system, which, in the case of a climate model, has $O(10^7)$ degree of freedom. If one considers a reduced phase space, the semi-group property, is unfortunately, lost. This has to do exactly with the fact that neglecting one or many variables leads to loss of markovianity in the system. This is carefully explained in Chekroun et al. 2015 For detailed yet compact description of these properties, please see section 4.1 in Tantet et al (2018), where the specific case of reduced order models in a climate model is discussed, and the criteria for choosing specific climate variables in order to (partly) reduce this issue is discussed. It is also discussed in my paper on J. Stat. Phys. you have cited (it is mentioned as a barrier for the method for computing the response).

Note that the loss of markovianity in a reduced space is the fundamental reasons why if one wants to construct parametrisation for unresolved processes (parametrisation is needed when we project our part of the phase space), one needs to include a memory term to take into account non-markovianity (Wouters and Lucarini 2012, Vissio and Lucarini 2017).

The second problem is that you patch together results from different models. Unfortunately, this is ill-defined in probabilistic terms also when computing multi-model averages, let alone when computing probability transitions. I know it is a common procedure used in many climate papers, but it is, indeed, wrong.

One may wonder why the results look promising. Well, also using $AR(n)$ fits one often gets good results in terms of hind-cast. But this does not mean that we have really achieved predictive skill.

Please take these comments as extremely constructive and not dismissive at all. As you can see, I am working on these issues and would be happy to have an off-line discussion with the authors.

Mickaël D. Chekroun, Honghu Liu, Shouting Wang, Stochastic Parameterizing Manifolds and Non-Markovian Reduced Equations, Springer 2015

Jeroen Wouters and Valerio Lucarini, Disentangling multi-level systems: averaging, correlations and memory, J. Stat. Mech. (2012) P03003

Vissio, G. and Lucarini, V., A proof of concept for scale-adaptive parametrizations: the case of the Lorenz '96 model. Q.J.R. Meteorol. Soc.. doi:10.1002/qj.3184 (2017)

Alexis Tantet, Valerio Lucarini, Frank Lunkeit, Henk A. Dijkstra, Crisis of the Chaotic Attractor of a Climate Model: A Transfer Operator Approach, arxiv:1507.02228, Nonlinearity, accepted (2018); text at <https://arxiv.org/pdf/1507.02228.pdf>

Reviewer #2 (Remarks to the Author):

This paper uses a statistical or machine learning simple prediction system to predict global mean sea surface and near surface temperature. The system is trained with 10 climate models, and the applied to observations. The evaluation metrics appear state of the art and the paper is generally well written and certainly noteworthy. It appears longer than necessary though.

Having a friendly competition between statistical and physical model-based prediction systems in climate research would certainly be useful for climate science. Nevertheless I have a few comments before I think this is ready for the journal:

a) the perfect model test is certainly very useful and appropriate. However, what really should be used to evaluate the predictions in the context of a possibly imperfect model with imperfect forcing, are imperfect model tests. That would be to omit a single model from the database, and predict its 3 or more ensemble members using the other 9, and also using the multimodel mean forced response based on the other 9. I think the GISS data are not spatially complete (although more infilled than HadCRUT) so it would also be nice to consider this in a prediction experiment, see next point. Without imperfect model predictions it will be harder to ascertain if the present good performance is partly down to luck given the small number of decadal samples.

b) it is not discussed in the paper how gaps in spatial data are handled. are the data considered approximations of global mean? in a perfect model test (and an imperfect one) it would be good to test if those gaps matter by omitting data from the models where the observations have gaps (ie missing gridpoints) and then form the global mean to see the significance of this issue

c) while the writing is generally excellent, aspects of it are too uncritical and adversarial. For example, I dont find it helpful to compare the computer time used between a physically based climate prediction and this statistical one without mentioning that the model based ones predicts the 4-D climate system not just global mean (you do this but much later)! So you put in less computer time, but you also get out less.

d) Results shown for the hiatus start in 1998 only if I capture this correctly although you have clearly done this for the entire record in another figure. could you please discuss a bit to what extent the start point matters here?

Specific points:

abstract: 22ms will certainly depend on the laptop and is unhelpful here: why not be a bit more subdued and say that prediction of global mean SST and SAT takes minutes on a laptop. and I dont see why I would want to predict global mean SAT on my phone? how would that be useful? (ok fascinating but useful?)

p 2 'can only occur' thats too narrow - better characterization of near term economic and emission predictions and natural forcing predictions will also help!

text on bottom of page 2 top of page 3 seems to preempt material later in the paper and seems out of place to me.

p. 3 end of page: 'after removing the part... forcing see method: the methods contain also almost nothing here. general attribution approaches use multilinear regression too so this is way not specific enough. I suspect you have done line fitting and removed the best match but how did you do this and what forcings go into it? I really dont like it when something gets bumped off to methods without being explained there!

end of p3: I have never encountered cK and dont think this is a useful addition to units. 0.1 K is

fine! Also, I would think the range of std devs for the instrumental period and the relative location of the observed residual in them is more helpful than the percent difference, and also I don't quite get where the 41% comes from - again not a useful metric. This should also be crosslinked to the AR5 box on the hiatus in chapter 9 which discusses this question (not just for the hiatus)

p 4, end of 2nd paragraph: I am not convinced by the skin temperature explanation here. There are good papers by Kevin Cowton explaining the role of SST vs SAT, but of course for SAT - but here for SST I am unconvinced as skin temperature I think is the uppermost top of the ocean which is not consistent with ship intake measurement, and the mixed layer should be well mixed so 10m is certainly no issue here. Unless you can support this I would reduce this speculation sharply

p 4 middle: I don't quite get the discussion on convergence to total density function could you eg show some figures in a supplement or describe this more clearly?

p. 7 top: you haven't returned before going to observations, right? maybe worth mentioning. (and if you have then I would be very sceptical of the paper!)

can you explain the meaning/consequences of reliability 2.3 vs 1?

p. 9 hiatus prediction discussion is a bit uncritical - if I read your predictions correctly, you do underestimate the amplitude, and that would be consistent with the possibility of further forcings. Citing a few review papers on the hiatus here would be helpful

p. 11 and discussion of final figure: you are predicting extremes of global SAT and SST not regional events - that should be made clearer in text and caption. You could relate to what extent global extreme T typically goes along with regional extremes.

citation 1: you cite this for attribution so citing the respective chapter is more appropriate I think

Figure 4 and 5 caption: The captions are very similar but poorly separated. eg 4 mentions hatching which only appears in 5, and I don't see the point of a flat reliability field plotted - why is it so flat? and what value is the pink hue that can't well be attributed to a very shallow colour scale?

Figure 7 caption: see above clarify anomalous event of global T.

Reviewer #1:

Let me (Valerio Lucarini) say that I honestly appreciate the style and goals of your paper (at many levels).

Thank you for this positive and supportive comment.

The more important reason has to do with the interpretation of the transfer operator in the case of reducing the system to just one variable - globally averaged surface temperature or sea temperature.

We feel that there is confusion about the method we used and how the Transfer Operator was applied. This is partially our fault since we do not use it in the usual way. This has now been fully clarified in the text.

The transfer operator is "well defined" if the system one is considering obeys the semigroup property. This means - broadly speaking - that the operator describing the transition of probabilities between 0 and time t can be written as the product of the evolution operator between 0 and t-\tau and the one between t-\tau and t, for any choice of \tau. This is the case if one works in the full space of the system, which, in the case of a climate model, has $O(10^7)$ degree of freedom. If one considers a reduced phase space, the semi-group property, is unfortunately, lost. This has to do exactly with the fact that neglecting one or many variables leads to loss of markovianity in the system. This is carefully explained in Chekroun et al. 2015 For detailed yet compact description of these properties, please see section 4.1 in Tantet et al (2018), where the specific case of reduced order models in a climate model is discussed, and the criteria for choosing specific climate variables in order to (partly) reduce this issue is discussed. It is also discussed in my paper on J. Stat. Phys. you have cited (it is mentioned as a barrier for the method for computing the response).

We fully agree with this description. However, our method differs from what you suggest and from the "more classical" use of Transfer Operators. We do not apply a single transfer operator several times successively to propagate forward in time the probability density function. Indeed, this would require the markovianity of the system, which is lost by considering only the global temperature (reducing the phase space to a single variable). Instead, we apply sequentially different transfer operators computed for different propagation times. This could be simply described as $T(1yr)=M1*T(0)$, $T(2 yr)=M2*T(0)$, ..., $T(n yr)=Mn*T(0)$. However, in contrast to the traditional Markov chain, in our method $M2$ is different from $M1*M1$. So, there is no need for markovianity.

Hence our method overcomes the difficulty described in the reviewer's comment above. Our method is computationally more expensive than the traditional Markov chain (since it requires the computation of transfer operators for each propagation time separately), and hence it slightly defeats the classical purpose of using a transfer operator, but it allows a more accurate description of the evolution of the probability density function, without the need for markovianity and bypassing its constraints. While the traditional Markov chain approach indeed would not be valid for our application (as we tested), through the adjustment we made, the use of Transfer Operators becomes perfectly valid.

This method was already fully described in the method part of the manuscript:

"The Transfer Operator is built by evaluating the number of trajectories [...] in each state and then evaluating the number of these trajectories ending-up in each possible state after a given transition time (τ). The ratio of these two numbers gives the probability from a trajectory in an initial state to end-up in a final state after a time τ (Fig. 2). The probability of state transition is repeated for $\tau=1, 2, 3, 4, 5, 6, 7, 8, 9$, and 10 years, leading to 10 Transfer Operators [...]."

This has been further clarified in the new version of the manuscript.

p.22: *“It is crucial to note that we do not apply a single Transfer Operator successively (e.g., applying the 1-yr Transfer Operator twice to get a 2-yr prediction). In contrast, we apply a range of Transfer Operators sequentially (i.e., applying the 1-yr Transfer Operator for 1-yr prediction, applying the 2-yr Transfer Operator for 2-yr, and so on). The essential difference with the traditional use of Transfer Operator (i.e., applied successively as a Markovian chain) is that we do not require that the 2-yr Transfer Operator is equal to applying the 1-yr Transfer Operator twice successively. So, we need to, and did, establish all different Transfer Operators for different lags separately and independently. This method indeed allows us to avoid the need for a Markovian chain (and its required properties), which, as a result, is not verified in our reduced single-variable space (Lucarini, 2016), and is probably not valid either. It should be emphasized that our method is closer to a conditional probabilistic prediction (where the condition is based on the current GMT or SST) than to the traditional use of Transfer Operators within a Markovian chain.”*

Note that the loss of markovianity in a reduced space is the fundamental reasons why if one wants to construct parametrisation for unresolved processes (parametrisation is needed when we project our part of the phase space), one needs to include a memory term to take into account non-markovianity (Wouters and Lucarini 2012, Vissio and Lucarini 2017).

We understand and agree with the point above. However as mentioned above, we do not assume markovianity of the system and, as a result do not need to include a memory-term to account for non-markovianity. One could argue that our different use of Transfer Operators implicitly includes such a memory-term by relaxing the demand that $M(2)=M(1)*M(1)$. When allowing for this extra degree of freedom, reducing the phase space to global temperature is enough to predict with good accuracy the global temperature (accuracy which surpasses state-of-the-art prediction system), as demonstrated in our manuscript. This method is at the core of our prediction system; and the good skill is a fundamental result of the paper.

The second problem is that you patch together results from different models. Unfortunately, this is ill-defined in probabilistic terms also when computing multi-model averages, let alone when computing probability transitions. I know it is a common procedure used in many climate papers, but it is, indeed, wrong.

We are not patching together results from different models. We are simply binning the results of all the models acknowledging that each model is one possible result. We do not feel that applying such post-processing statistic is ill-defined (albeit affected by the well-known smoothing that any binning implies). We agree that it would be different if we were using a single transfer operator successively. In that case, some probabilities would come from a trajectory from model 1, followed by trajectory from model 2 (for instance), which would lead to a fundamental inconsistency (there is no model that is actually able to create such a “Frankenstein”-artificial trajectory). But, as mentioned in the response to your first/main comment, we are not doing that.

In our view, the implicit assumption we make, a-posteriori verified by the results, is that each single model has a biased probability distribution. By binning a large enough number of models, the bias in probability distribution is reduced, and now becomes a multi-model averaged bias, which is smaller than any single-model bias, as it only contains the bias that is common to all models.

Please see below how we modified the manuscript regarding Specific Points on the statistical methodology.

One may wonder why the results look promising. Well, also using AR(n) fits one often gets good results in terms of hind-cast. But this does not mean that we have really achieved predictive skill.

Indeed we have demonstrated good prediction skills of GMT and SST (Fig. 5), as well as their evolution (Fig. 6). We hope, that the above clarification on our method sheds some light on the reason why our

method achieved good skill. We have indeed tested applying a single transfer operator (trained with a 1 year transition time) successively. As expected we did not get any skill (probably due to the lost of markovianity). However our method of training transfer operators for each transition times do show significant predictive skills.

Redacted

Reviewer #2:

This paper uses a statistical or machine learning simple prediction system to predict global mean sea surface and near surface temperature. The system is trained with 10 climate models, and the applied to observations. The evaluation metrics appear state of the art and the paper is generally well written and certainly noteworthy.

Thank you for this supportive comment.

Having a friendly competition between statistical and physical model-based prediction systems in climate research would certainly be useful for climate science.

We agree with the reviewer on the usefulness of friendly competition. We also would like to stress that our predictive system is “physical model-based” (not statistical). We use statistics to build the probabilistic prediction, but this statistics is based on “physical model-based”. We feel this is the fundamental reason of the skill of our system.

a) the perfect model test is certainly very useful and appropriate. However, what really should be used to evaluate the predictions in the context of a possibly imperfect model with imperfect forcing, are imperfect model tests. That would be to omit a single model from the database, and predict its 3 or more ensemble members using the other 9, and also using the multimodel mean forced response based on the other 9.

We have done this set of experiments. The resulting skills are summarized in the Figure below. This shows that the imperfect model approach has equivalent skill as the perfect model (with reliability ~1 and an slight decrease of Coefficient of Determination bound to 0.01). This further shows the robustness of the method.

Caption: As Figure 4 but for the imperfect model approach – Skill differences (e-f) are imperfect vs perfect approach.

We have modified the text to include this new result.

p.7: "To further test the predictive skill and reliability of PROCAS^T we have assessed them in an imperfect model approach (i.e., removing outputs of one model from the Transfer Operator computation and using them as pseudo-observations). We find that PROCAS^T is still able to perform at the same level of accuracy than within the perfect model approach with a slight decrease of the coefficient of determination of less than 0.01 for all lags and averaging times tested."

I think the GISS data are not spatially complete (although more infilled than HadCRUT) so it would also be nice to consider this in a prediction experiment, see next point. Without imperfect model predictions it will be harder to ascertain if the present good performance is partly down to luck given the small number of decadal samples.

We have used 136 starting dates and applied hindcasts for 1 to 10-yr averages (which are, in the case of PROCAS^T, independent). This leaves 1,360 independent decadal hindcasts. We feel that it is a large enough sample to test the robustness of our prediction... In particular, the extremely smooth statistics of our skill in its operational mode (Fig.5) further suggest the accuracy of the computed skill.

b) it is not discussed in the paper how gaps in spatial data are handled. are the data considered approximations of global mean?

Indeed we have treated it as an approximation of global mean. This is now fully acknowledged in the text.

p.24: "GMT and SST are computed as spatial averages (NASA GISS temperature record for GMT; the NOAA ERSSTv5 record for SST), where spatial gaps in the data are ignored. The percentage of missing data (important before 1958) does not show any impact on the prediction skill. The internal variations in GMT and SST in the observational record are computed as the residual after having removed the part that can be attributed to external forcing."

in a perfect model test (and an imperfect one) it would be good to test if those gaps matter by omitting data from the models where the observations have gaps (ie missing gridpoints) and then form the global mean to see the significance of this issue

We agree that it might be problematic. To test it, rather than recomputing the prediction with artificial gaps in the model data, we have plotted the percentage of missing data for the starting year against the hindcast error, within the fully operational framework (Figure below).

This clearly shows that missing data is significant until 1958 with missing values from 32% to 15%. After 1958 the good coverage leads to a quite stable level of missing values of less than 2% (Figure below). (Note that we have been extremely conservative to compute the missing values, since a single months of missing values is considered as a missing values for the entire year.) On the other hand, the prediction error does not show significant difference between pre- and post-1958, with an average error of 8 mK and of 8 mK; and a standard deviation of 9 mK and of 10 mK respectively (Figure below). Hence no statistical difference can be determined between pre- and post-1958. Finally we plotted the prediction error with the percentage of missing data for each year to evaluate any possible relationship (Figure below). We did not find any clear relation.

Caption: (top) Percentage of missing data from GISS as a function of time. (middle) 1-yr lag prediction error of PROCAST in its fully operational mode. The vertical black line represents year 1958 when missing data becomes weak (less than 2%); thick (thin) red and blue lines represent time-mean (time-mean + one standard deviation) for the pre- and post-1958 periods, respectively. (bottom) 1-yr lag prediction error of PROCAST (in its fully operational mode) as a function of the percentage of missing data from GISS. Red and blue crosses represent time-mean percentage of missing data and of 1-yr lag prediction error.

We have added a comment on this point in the text.

p.24: "GMT and SST are computed as spatial averages (NASA GISS temperature record for GMT; the NOAA ERSSTv5 record for SST), where spatial gaps in the data are ignored. The percentage of missing data (important before 1958) does not show any impact on the prediction skill. The internal variations in GMT and SST in the observational record are computed as the residual after having removed the part that can be attributed to external forcing."

c) while the writing is generally excellent, aspects of it are too uncritical and adversarial. For example, I dont find it helpful to compare the computer time used between a physically based climate prediction and this statistical one without mentioning that the model based ones predicts the 4-D climate system not just global mean (you do this but much later)! So you put in less computer time, but you also get out less.

We had a thorough read to the paper and have toned down any claim that was too uncritical or too adversarial.

Regarding discussion between the relative benefit of DePreSys3 and PROCAST. The discussion about the single variable prediction (PROCAST) and the full state variable (DePreSys3) is done in the sentence directly following computer time efficiency. We do not feel it is so late. This is the only time we compare computer times.

d) Results shown for the hiatus start in 1998 only if I capture this correctly although you have clearly done this for the entire record in another figure. could you please discuss a bit to what extent the start point matters here?

We mention this hiatus has an emblematic example (i.e., post-1998 hiatus). As mentioned by the reviewer, this decadal prediction has been done using each year as the starting year. Obviously, the start date matters for the skill (as shown below and summarized in Fig. 5).

We have tested a starting year in 2002 (see figure below). The Correlation Coefficient is better whereas the R-square is more moderated than for using 1998 as starting year. PROCASST still predicts a cooling of the anomaly partially compensating the forced warming over this period.

Caption: As Figure 6 but for a starting day in 2002.

The dependency of the skill to starting year is now fully acknowledged in the text.

p.10: "Other starting dates have been tested (such as 2002) and always allow PROCASST to capture the long-term hiatus, but to a lesser degree the exact interannual variation of the decades."

Specific points:

abstract: 22ms will certainly depend on the laptop and is unhelpful here: why not be a bit more subdued and say that prediction of global mean SST and SAT takes minutes on a laptop. and I dont see why I would want to predict global mean SAT on my phone? how would that be useful? (ok fascinating but useful?)

We agree and have changed the sentence to be less specific. However "minutes" seems way off our actual efficiency (by a factor of a thousand at least).

p.1: "The extreme numerical efficiency of the method (a few hundredth of a second for a decadal prediction on a laptop)"

and I dont see why I would want to predict global mean SAT on my phone? how would that be useful? (ok fascinating but useful?)

We hope that we will soon be able to do prediction of other variables and timescales hence "opens the possibility". This would be quite neat if one can make predictions of what he/she wants by carrying his/her own prediction system his/her pocket.

p 2 'can only occur' thats too narrow - better characterization of near term economic and emission predictions and natural forcing predictions will also help!

We agree that it was too restrictive. We changed the sentence.

p.2: "[...], further improvement of climate predictions will mainly occur through better, more accurate predictions of the internal variability."

text on bottom of page 2 top of page 3 seems to preempt material later in the paper and seems out of place to me.

It was a summary of the method and result. It is probably not needed for a letter. We shortened it.

p. 3 end of page: 'after removing the part... forcing see method: the methods contain also almost nothing here. general attribution approaches use multilinear regression too so this is way not specific enough. I suspect you have done line fitting and removed the best match but how did you do this and what forcings go into it? I really dont like it when something gets bumped off to methods without being explained there!

We agree the method description was too general and superficial and mainly referring to published studies. We have clarified this point in the method section.

p.24-25: "For removing the part attributed to external forcing from the timeseries of SAT and SST a multiple linear regression analysis is performed, following ref [2]. The approach assumes that globally averaged temperature responds linearly, with some lag, to the various forcing agents. After removing the average value (we are only interested in anomalies), we can write:

$$T(t) = \varepsilon + \sum_{i=1}^n a_i F_i(t-l_i),$$

where T is the total GMT or SST, n is the number of forcings considered (i.e., three: anthropogenic forcing, solar forcing, and volcanoes), a_i are the regression coefficients, F_i are the forcing timeseries, l_i is the lag by which temperature responds to the forcing, and ε is the residual. The forcing time series are taken from the CMIP5 historical plus RCP4.5 (after 2005 till present) forcing dataset. We refer to ref [2] for more details and figures of how the regression performs."

end of p3: I have never encountered cK and dont think this is a useful addition to units. 0.1 K is fine!

We agree. We have modified the text.

Also, i would think the range of std devs for the instrumental period and the relative location of the observed residual in them is more helpful than the percent difference, and also I dont quite get where the 41% comes from - again not a useful metric. THIS should also be crosslinked to the AR5 box on the hiatus in chapter 9 which discusses this quesiton (not just for the hiatus)

We feel that there is confusion in the goal of this sentence. What we mean is that the spread of the total density function is different in the observations than in the model outputs. This difference is small for GMT but important of the order of 41% for SST. Hence for SST we applied a renormalization of the density function so that the spread of the modeled values fits the observed one.

We have clarified this point:

p.3-4: *"The CMIP5 GMT and SST anomalies consist of centered distributions with a standard deviation of the annual mean of 0.1 K and 0.07 K, respectively. For GMT, the modeled standard deviation is slightly weaker than the observed one (0.12 K), but remains in good agreement: less than 9% of relative difference. On the other hand, for SST, the standard deviation of the distribution in the CMIP5 is significantly weaker than in the observations (0.13 K), with a relative difference of 43%. Hence for SST, the modeled distribution is renormalized to fit the standard deviation of the observations."*

p 4, end of 2nd paragraph: I am not convinced by the skin temperature explanation here. There are good papers by Kevin Cowton explaining the role of SST vs SAT, but of course for SAT - but here for SST I am unconvinced as skin temperature I think is the uppermost top of the ocean which is not consistent with ship intake measurement, and the mixed layer should be well mixed so 10m is certainly no issue here. Unless you can support this I would reduce this speculation sharply

We agree that the difference might be more complex than we suggested. We have removed the sentence to avoid confusion.

p 4 middle: I don't quite get the discussion on convergence to total density function could you eg show some figures in a supplement or describe this more clearly?

Fig.3 gives an example of the convergence of the density function toward the "climatological"/total density function. We are now referring more explicitly to Fig. 3 when discussing the asymptotic convergence.

p.4: *"This shows that for all possible initial conditions the probability density function slowly converges to the total density distribution with a timescale of ~10 yr (see Fig. 3 for an example of the asymptotic convergence)."*

p. 7 top: you haven't returned before going to observations, right? maybe worth mentioning. (and if you have then I would be very sceptical of the paper!)

No there is absolutely no "retuning" done. This is the exact same statistics that are used. This is fully mentioned in the paper now.

p.7: *"After having tested PROCAST in a perfect model setting, we now test the exact same system with real observations. (Note that no retuning before going to observations has been applied.)"*

can you explain the meaning/consequences of reliability 2.3 vs 1?

This was explained in the method. We have added specific comments in the main text for completeness.

p. 8-9: *"More specifically, PROCAST reliability for annual GMT is almost perfect with an averaged value for hindcast lags from 1 to 5 yr of 1 (prediction spread is as big as the prediction error on average), whereas it is 2.3 for DePreSys3 (prediction spread is 2.3 times as small as the prediction error on average). This relatively weak reliability of DePreSys3 is a sign of the under-dispersion of the ensemble in comparison with its prediction error and of an over-confident prediction system."*

p. 9 hiatus prediction discussion is a bit uncritical - if I read your predictions correctly, you do underestimate the amplitude, and that would be consistent with the possibility of further forcings. Citing a few review papers on the hiatus here would be helpful

We generally agree with this comment. The actual amplitude of the hiatus was correctly predicted in a probabilistic sense staying well within +/- standard deviation of the probabilistic prediction. The goal

was not to discuss hiatus, but to give an example. We have added references and give a more critical view of our results.

p.9-10: "Despite some error in its exact intensity (especially when focusing of mean prediction) or the details of its annual variations, this shows that an event such as the post-1998 hiatus could not have been missed using PROCAST (especially when acknowledging the predictive spread). In particular our probabilistic forecast framework shows that a decade-long hiatus was always a likely outcome (always well within 1 standard deviation of our prediction), even if not the most likely, especially after 7 yr. Because the amplitude is somewhat lower than observed, it would be consistent if a small part of the hiatus was indeed caused by external forcing, although the main part would be due to internal variability¹⁸⁻²¹."

p. 11 and discussion of final figure: you are predicting extremes of global SAT and SST not regional events - that should be made clearer in text and caption. You could relate to what extent global extreme T typically goes along with regional extremes.

We agree we are only discussing global prediction. The two metrics we used since the beginning of the study. This has been clarified in the text by adding the words global when necessary.

We are not aware of any studies relating the extreme GMT with regional extremes, except for the subset of ENSO related GMT change.

citation 1: you cite this for attribution so citing the respective chapter is more appropriate I think

Done

Figure 4 and 5 caption: THE captions are very similar but poorly separated. eg 4 mentions hatching which only appears in 5, and I dont see the point of a flat reliability field plotted - why is it so flat? and what value is the pink hue that cant well be attributed to a very shallow colourscale?

We agree that some results of Fig.4 could have been better explained. We choose to use the same layout and color scales between Fig.4 and 5 to help comparison (i.e., comparison between perfect model approach and hindcast in operational mode). The absence of hatching in Fig.4 is because the model surpasses persistence for all hindcast lags and averaging timescales. The flat pink color is because the Reliability is perfect and so equal to 1 for all hindcast lags and averaging timescales (as expected in perfect model approach). Both properties were mentioned in the caption: "Note the better skill than persistence (no hatched region) and the good reliability close to 1 for all hindcast lags and averaging times." We have clarified that in the new version of the caption of Fig.4.

Caption Fig. 4: "Note the absence of hatched region in a and b denoting the better skill than persistence for all hindcast lags and averaging times. Also, note the flat pink colour in c and d corresponding to a good reliability close to 1 for all hindcast lags and averaging times, as expected in a perfect model approach."

Figure 7 caption: see above clarify anomalous event of global T.

Done

REVIEWERS' COMMENTS:

Reviewer #1 (Remarks to the Author):

Dear authors,

I wish to say that the quality of your manuscript has substantially improved compared the previous version. I encourage publication of the paper but I kindly ask the authors to keep into account the comments below:

1) At page 3 and 4, some information about the limitations is the methodology caused by using a severely projected phase space should be briefly discussed (the current explanation in the appendix is ok). Otherwise, the reader would be slightly misled.

2) I really do not understand this sentence

"Here, the climate state is evaluated through the 1-dimensional phase space defined by either GMT or SST, whereas the state transitions are based on the evolution of the respective metric in the CMIP5 database"

3) Assumption 4 as discussed in the appendix: the entire IPCC report on extremes (which has its own limitations) discusses changes in extremes as due to combined changes of the mean and of the variability of the distribution. This is - at many levels - a grossly simplified view on the problem. The authors should clarify that their assumption, which neglects dealing with changes in the natural variability, might lead to low skill in predicting the probability of occurrence of extremes, which is instead something they focus on also in the abstract.

4) I honestly believe that, when mentioning the problem of markovianity (or lack of) in your estimates of the transfer operator, and, in fact, to provide support to your approach, you should refer to the recently published paper by Tantet et al Nonlinearity 2018

<http://iopscience.iop.org/article/10.1088/1361-6544/aaaf42/pdf>

where this is discussed in detail, when looking at tipping points and analysing decay of correlations.

5) You say that your way of performing climate prediction in the next few years is very cheap and efficient. But please do not forget that you are using datasets that have been produced with incredibly expensive numerical exercises!

All the best,

Valerio Lucarini

Reviewer #2 (Remarks to the Author):

The authors have addressed my comments well and my major questions have been resolved. I am happy for the ms to go forward, as it is very timely and delaying it longer would miss opportunities. I have a few suggestions for the authors to decide on: title: its interesting but when I saw the title I wasnt sure it was the paper I was thinking of! maybe have at least the word 'predicting' in title?

abstract: I am still not convinced the world needs mobile predictable global SSTs or SATs. I think it would be more useful here to point at many impacts of climate change

scaling with global temperature and hence make a link to impacts.

The section about predictability vs reliability ending in the middle of page 6 is a bit verbose. I recommend cutting it a bit by removing duplications.

p. 10: its interesting that using the 'best' models doesnt give you better predictions but could you here briefly explain what you mean by 'best'? its not clear immediately.

Reviewer #1:

1) At page 3 and 4, some information about the limitations is the methodology caused by using a severely projected phase space should be briefly discussed (the current explanation in the appendix is ok). Otherwise, the reader would be slightly misled.

Done.

p.4: "In particular, the severe truncation of the phase space to a single variable implies that different climate states with equivalent GMT or SST are all aggregated in the probabilistic approach of the Transfer Operators."

2) I really do not understand this sentence

"Here, the climate state is evaluated through the 1-dimensional phase space defined by either GMT or SST, whereas the state transitions are based on the evolution of the respective metric in the CMIP5 database"

We have rephrased the sentence:

p.3: "Here, the climate state is evaluated through the 1-dimensional phase space defined by either GMT or SST, whereas the state transitions are based on GMT or SST evolutions simulated by climate models from the CMIP5 database (see Methods for further details)."

3) Assumption 4 as discussed in the appendix: the entire IPCC report on extremes (which has its own limitations) discusses changes in extremes as due to combined changes of the mean and of the variability of the distribution. This is - at many levels - a grossly simplified view on the problem. The authors should clarify that their assumption, which neglects dealing with changes in the natural variability, might lead to low skill in predicting the probability of occurrence of extremes, which is instead something they focus on also in the abstract.

We have now mentioned this shortcoming.

p.23-24: "Assumption 4 is also not correct, although generally applied in climate science. The general assumption is that the perturbation implied by global warming is too small to fundamentally change the anomalies, or climate variability (i.e., anomalies and perturbation to the background do not affect each other). This might be problematic when studying extreme events, whose occurrences might change in a changing climate. Hence, this assumption is questionable for certain variables that are subject to order one changes under climate change, such as, e.g. the Atlantic Meridional Overturning Circulation, or extremes in the tail of a distribution. However, there is no indication that the assumption is not approximately valid for GMT, whose variability is for a large part dominated by El Niño-Southern Oscillation and the Interdecadal Pacific Oscillation. So, despite these various assumptions, PROCAS is skillful for interannual GMT and (globally averaged) SST prediction, proving that these assumptions although not strictly true are reasonable in the context of our study, because they apply in most cases of variation in GMT."

4) I honestly believe that, when mentioning the problem of markovianity (or lack of) in your estimates of the transfer operator, and, in fact, to provide support to your approach, you should refer to the recently published paper by Tantet et al Nonlinearity 2018

<http://iopscience.iop.org/article/10.1088/1361-6544/aaaf42/pdf>

where this is discussed in detail, when looking at tipping points and analysing decay of correlations.

We have added this reference in the discussion on p.22.

5) You say that your way of performing climate prediction in the next few years is very cheap and efficient. But please do not forget that you are using datasets that have been produced with incredibly expensive numerical exercises!

p.9: "This difference in numerical cost has to be put into perspective, though. PROCAST takes advantage of the freely available CMIP5 database, which is an incredibly expensive numerical exercise of the worldwide climate science community. Also, unlike PROCAST, DePreSys3 is not specifically trained for a single variable prediction, so that the entire climate state is predicted in one forecast. This is obviously beneficial."

Reviewer #2:

title: its interesting but when I saw the title I wasnt sure it was the paper I was thinking of! maybe have at least the word 'predicting' in title?

We agree with this comment and have modify the title:

Title: "A Novel Probabilistic Forecast System Predicting Anomalously Warm 2018-2022 Reinforcing the Long-Term Global Warming Trend"

abstract: I am still not convinced the world needs mobile predictable global SSTs or SATs. I think it would be more useful here to point at many impacts of climate change scaling with global temperature and hence make a link to impacts.

We agree that the impact of GMT changes is significant and have added this to the conclusion.

p.15: "This also opens the possibility of giving access to climate forecast and possible subsequent impacts that scale with GMT to a wider scientific community (without the need for super-computer) and to the general public by running a simple application on a personal portable device."

The section about predictability vs reliability ending in the middle of page 6 is a bit verbose. I recommend cutting it a bit by removing duplications.

We have shortened the text.

p5-6: "To estimate the validity of our probabilistic predictions we use two different measures: the coefficient of determination – R^2 , which shows the skill of the mean prediction; and the Reliability, which measures the accuracy of the spread in the prediction. These two measures can be mathematically expressed as:

**** Eq 1a ***,*

**** Eq 1b ***,*

where t is time, i is the possibility or state index, $o(t)$ is the observation and x_i are the predicted possibilities with probability $p_i(t)$. The bar denotes an average over time or possibilities depending on the superscript. (Our equation of the Reliability is an extension for non-stationary statistic of the previously suggested definition¹³.) The coefficient of determination, when multiplied by 100, gives the percentage of variance of the observation explained by the prediction. Since the system is chaotic (there is a degree of uncertainty around the mean prediction), it is expected that the prediction cannot represent the observation perfectly, even if the model represents perfectly reality. Hence the reliability measures the accuracy of this prediction error. When a reliable prediction has large skill (~ 1) we expect the prediction uncertainty to be small. On the other hand, when a reliable prediction system has low skill (~ 0) we expect the prediction uncertainty to be as big as the observed variance. In this context, and regardless of its skill, a reliable prediction system always needs to have a Reliability close to 1. Hence, despite that a high value of R^2 is preferable for a skillful prediction, the reliability is arguably more important to estimate the usefulness of the prediction system. Indeed a reliable prediction system can be used for probabilistic forecasts and risk assessments, even if it has low skill^{14,15}."

p. 10: its interesting that using the 'best' models doesnt give you better predictions but could you here briefly explain what you mean by 'best'? its not clear immediately.

We have modified the text.

p9: "It is also interesting to note that skill and reliability are improved by the addition of information from more models rather than by selecting a subset of the best models (i.e., models giving the best skill when used alone to train the Transfer Operator)."